# Reference Genes Screening and Gene Expression Patterns Analysis Involved in Gelsenicine Biosynthesis under Different Hormone Treatments in *Gelsemium elegans*

**DOI:** 10.3390/ijms242115973

**Published:** 2023-11-04

**Authors:** Yao Zhang, Detian Mu, Liya Wang, Xujun Wang, Iain W. Wilson, Wenqiang Chen, Jinghan Wang, Zhaoying Liu, Deyou Qiu, Qi Tang

**Affiliations:** 1College of Horticulture, National Research Center of Engineering Technology for Utilization of Botanical Functional Ingredients, Hunan Agricultural University, Changsha 410128, China; ningzby@163.com (Y.Z.); mudetian12580@163.com (D.M.); wly021006@outlook.com (L.W.); 15958699677@163.com (W.C.); 2Hunan Academy of Forestry, Changsha 410018, China; 3CSIRO Agriculture and Food, Canberra, ACT 2601, Australia; iain.wilson@csiro.au; 4College of Forestry, Central South University of Forestry and Technology, Changsha 410004, China; m15200830187@163.com; 5College of Veterinary Medicine, Hunan Agricultural University, Changsha 410128, China; liu_zhaoying@hunau.edu.cn; 6State Key Laboratory of Tree Genetics and Breeding, Research Institute of Forestry, Chinese Academy of Forestry, Beijing 100091, China; qiudy@caf.ac.cn

**Keywords:** RT-qPCR, reference gene, *Gelsemium elegans*, normalization, hormone treatment, co-expression, gelsenicine

## Abstract

Reverse transcription quantitative polymerase chain reaction (RT-qPCR) is an accurate method for quantifying gene expression levels. Choosing appropriate reference genes to normalize the data is essential for reducing errors. *Gelsemium elegans* is a highly poisonous but important medicinal plant used for analgesic and anti-swelling purposes. Gelsenicine is one of the vital active ingredients, and its biosynthesis pathway remains to be determined. In this study, *G. elegans* leaf tissue with and without the application of one of four hormones (SA, MeJA, ETH, and ABA) known to affect gelsenicine synthesis, was analyzed using ten candidate reference genes. The gene stability was evaluated using GeNorm, NormFinder, BestKeeper, ∆CT, and RefFinder. The results showed that the optimal stable reference genes varied among the different treatments and that at least two reference genes were required for accurate quantification. The expression patterns of 15 genes related to the gelsenicine upstream biosynthesis pathway was determined by RT-qPCR using the relevant reference genes identified. Three genes *8-HGO*, *LAMT*, and *STR*, were found to have a strong correlation with the amount of gelsenicine measured in the different samples. This research is the first study to examine the reference genes of *G. elegans* under different hormone treatments and will be useful for future molecular analyses of this medically important plant species.

## 1. Introduction

Reverse transcription quantitative polymerase chain reaction (RT-qPCR) is now widely used in molecular biology research as a reliable means of measuring gene expression levels, which is highly sensitive, specific, accurate, and rapid [1,2,3]. However, RT-qPCR results are susceptible to interference by reference genes. Reference genes are often required to normalize the data for the target gene to minimize errors [4,5]. However, a large number of studies have revealed that these genes might not be stably expressed among all species, or under specific experimental conditions. For example, different stable reference genes for *Luffa cylindrica* [6], *Cryptomeria fortunei* [7], *Allium sativum* L. [8] have been identified under different conditions. Therefore, it is essential to select the best reference genes based on various factors to ensure accurate RT-qPCR results.

A number of analyses such as GeNorm, NormFinder, BestKeeper, ∆CT, and RefFinder are commonly used to estimate the stability of candidate reference genes [9,10,11,12,13,14]. Among these, GeNorm is used to rank potential reference genes by the mean pairwise change in expression of a potential reference gene compared to every other gene in the set, and to assess the variability between genes [15]. NormFinder compares candidate reference genes in pairs within and between groups, and BestKeeper uses an analytical procedure based on the mean cycle threshold (Ct) values of candidate reference genes [16,17,18]. The ∆CT method compares the relative expression levels of pairs of genes in each sample [19]. RefFinder is used to combine the above algorithms to obtain a final result [20]. Currently, suitable reference genes that can be used for normalizing RT-qPCR gene expression analyses in *Gelsemium elegans* under different experimental conditions have not been reported.

*G. elegans* is an evergreen woody vine of the genus *Gelsemium* in the family *Loganiaceae*, mainly found in China, and southeastern Asia [21]. The whole plant is highly poisonous and is a traditional Chinese herb remedy for many symptoms [22]. Its roots are rich in monoterpene indole alkaloids (MIAs), which have demonstrated excellent anti-inflammatory, anti-tumor, anti-anxiety, immunomodulatory, and analgesic effects with promising medical applications [23,24,25,26,27,28]. Gelsenicine is an ingredient of MIAs and is the most toxic alkaloid of *G. elegans.* It has many pharmacological effects [29]. Although the biosynthesis of gelsenicine is still poorly understood, it has been found that its upstream biosynthesis pathways are closely related to that found in the iridoids and indole pathways (Figure 1) [30]. The iridoid pathway starts with the condensation of isopentenyl diphosphate (IPP) and dimethylallyl diphosphate (DMAPP) to form geranyl diphosphate (GPP). Its dephosphorylation by the action of geraniol synthase (GES) leads to the formation of geraniol, followed by various enzymic reactions (ten steps) to form secologanin. The indole pathway begins with chorismite that is catalyzed by anthranilate synthase (AS) to form anthranilate, followed by various enzymic reactions (six steps) to give tryptamine. Finally, strictosidin synthase (STR) catalyzes the formation of strictosidine from secologanin and tryptamine [31,32,33,34]. Gelsemium alkaloids are then obtained. However, the molecular understanding of gelsemium alkaloids and the associated biosynthetic pathways have been little studied in detail. 

Several studies have shown that exogenous hormones such as salicylic acid (SA), methyl jasmonate (MeJA), ethylene (ETH), and abscisic acid (ABA), induced or otherwise, alter the biosynthesis of secondary metabolites [35,36,37,38]. Catharanthine in *Catharanthus roseus* [39], isoquinoline alkaloids in *Dendrobium officinale* PLBs [40], camptothecin (CPT) in *Camptotheca acuminata* [41], have been found in response to hormones on biosynthetic pathways. Nevertheless, there is still a gap in research on the effects of exogenous plant hormones on the MIA biosynthesis of *G. elegans*. Therefore, in order to further understand the biosynthesis pathway to gelsemium alkaloids, identifying the expression patterns of pathway-related genes in response to different environmental stimuli is an important step.

In this study, ten candidate reference genes were selected based on the genomic data [42] and then assessed for stability of gene expression under SA, MeJA, ETH, and ABA treatments. Five algorithms were used for the systematic screening of optimal reference genes. Finally, *STR*, an important target gene in MIA upstream biosynthetic pathways, was used to validate the reference genes identified. In addition, we also screened 14 genes potentially involved in the gelsenicine upstream pathways based on a “genome + transcriptome + metabolome” co-expression analysis and analyzed their expression patterns. This work identified the best reference genes for RT-qPCR analysis of *G. elegans* under the different hormone treatments performed and illustrates the expression patterns of biosynthesis-related genes. This provides a foundation for further molecular studies of the MIA biosynthesis in *G. elegans*.

## 2. Results

### 2.1. The Content of Gelsenicine under Four Hormone Treatments

The chromatograms of the standard solution and sample solution are shown in Appendix A. The different hormone treatments of *G. elegans* leaves significantly affected the content of gelsenicine over a 48 h period compared to the control (Figure 2). After 48 h, the content of gelsenicine after treatment with SA (0.176 mg·g^−1^, 51.33%) and MeJA (0.205 mg·g^−1^, 59.93%) showed highly significant decreases (*p* < 0.01); ABA (0.381 mg·g^−1^, 111.28%) showed a significant increase (*p* < 0.05) and there was no significant change observed with ETH. Thus, addition of SA and MeJA had a very positive impact on the content of gelsenicine in *G. elegans* leaves.

### 2.2. Amplification Specificity, Efficiency, and Expression Profile Analysis of Reference Genes in G. elegans 

The specificity of the primers was measured by the RT-qPCR melting curve, and the results showed that all reference genes had a distinct single peak, indicating good specificity of all primers (Appendix A). Also, the amplification efficiencies of the 10 candidate genes ranged from 97.9% to 110.1%, with the lowest and highest values for *18S* and *EF1-α*, respectively (Table 1). In addition, standard curves, calculated using five-fold dilution of cDNA, had correlation coefficients (*R*^2^) greater than 0.99 for all candidate genes, which indicated a good linear relationship of the data obtained by RT-qPCR.

The expression levels of the 10 candidate genes can be gauged by calculating the RT-qPCR cycle threshold (Ct) values, assuming reaction efficiencies are comparable. A lower Ct value is associated with a higher gene expression abundance. The data showed that the expression patterns of these reference genes were very diverse (Figure 3), with Ct values ranging between 16.97–28.10. The highest expression level candidate was *EF1-α* (18.35 ± 0.99), and the lowest expression level candidate normalization gene was *SAND* (25.78 ± 1.07). Furthermore, *GAPDH* (19.26 ± 1.39) showed the largest variation in expression level, whereas *PP2A* (22.61 ± 0.50) showed a small range of variation. The relatively small Ct ranges of *UBC*, *PP2A,* and *Actin* as compared to the other candidate genes suggest that their expression is likely to be more stable across the samples analyzed.

### 2.3. Stability Analysis of Candidate Reference Genes

Five tools (GeNorm, NormFinder, BestKeeper, ΔCT, and RefFinder) were used to assess the expression stability of ten candidate genes for *G. elegans* leaves after hormone treatment (Table 2). The experimental datasets were divided into six groups, Control (0 h), SA, MeJA, ETH, ABA, and Total group (all samples), and all genes in these six groups were analyzed independently.

#### 2.3.1. GeNorm Analysis of Reference Genes in *G. elegans*

GeNorm assessed gene stability by calculating M-values. Smaller M-values (M) indicate better stability. The M-value for each gene was less than 1.5 in all groups (Figure 4). The results showed that *CDC25* and *EF1-α* had the highest stability both in the groups Control and SA, while *Actin* and *EF1-α* were the two most stable genes for groups MeJA and ABA. For ETH and Total group, *CDC25* and *PP2A*, *18S* and *UBC* were the two most stable genes, respectively. In contrast, *18S* was the least stable gene for groups Control and SA, while ranked the last for its stability in groups MeJA and Total. In addition, *EF1-α* and *GAPDH* were the bottom gene in groups ETH and ABA. Therefore, there are obvious differences observed in gene expression stability for different treatments, and follow-up studies were required to identify the most suitable reference genes.

GeNorm analysis also requires the pairwise variation (V) values, by which V_*n*/*n*+1_ determines the optimum number of reference genes to be selected. The V_2/3_ values in four treatments and Control group were less than 0.15 (Figure 5), which means that at least two genes are required to improve the reliability and accuracy of the results when selecting normalized reference genes under these five treatments. In contrast, the total group required at least three reference genes for data processing.

#### 2.3.2. NormFinder Analysis of Reference Genes in *G. elegans*

NormFinder was used to assess gene expression stability by analyzing intra- and inter-group variation. The results differ a little from GeNorm (Table 2) and showed that *TUA* and *GAPDH*, *EF1-α*, and *GAPDH*, *UBC* and *TUA*, *Actin* and *PP2A*, *SAND* and *TUB*, *Actin* and *UBC*, were the two most stable genes in groups Control, SA, MeJA, ETH, ABA, and Total, respectively. In contrast, the least stable genes were identical to GeNorm in all groups. 

#### 2.3.3. BestKeeper Analysis of Reference Genes in *G. elegans*

BestKeeper judges stability by using CV and standard deviation (SD, usually SD < 1) values, with smaller values indicating better stability of candidate reference genes. As shown in Table 2, the stabilities of *Actin* and *UBC* were the highest under the Control group. The SD values of all reference genes were low (<0.49) across the SA treatment group, with the two most stable genes being *GAPDH* and *CDC25*. *UBC* and *GAPDH* showed the highest stability under MeJA treatment. The stability of ETH treatment was best with *Actin* and *PP2A*. Under ABA treatment, *PP2A* and *CDC25* were shown as the two most stable genes. From the Total group, the two genes that had the highest stability were *PP2A* and *UBC*. Interestingly, the bottom ranked genes were consistent with GeNorm and NormFinder except for the ETH and Total groups, with *UBC* and *GAPDH* being the most unstable genes in these two groups, respectively. In addition, the SD of the candidate genes under most treatments was <1.

#### 2.3.4. ΔCT Method of Reference Genes in *G. elegans*

The ΔCT method assesses the stability based on the SD, and the SD value is inversely proportional to the stability using the raw Ct value with while the minimum SD corresponds to the most stable gene (Table 2). The results showed that, the best two stable genes were *TUA* and *GAPDH*, *EF1-α* and *GAPDH*, *UBC* and *TUA*, *Actin* and *PP2A*, *Actin and SAND*, *Actin* and *UBC* for groups Control, SA, MeJA, ETH, ABA, and Total, respectively, which were similar to the results of NormFinder. In summary, the results of this method were very similar to those assessed by NormFinder.

#### 2.3.5. RefFinder Analysis of Reference Genes in *G. elegans*

The results of the four algorithms mentioned above varied, and RefFinder was used to derive the geometric mean of the results from them (Table 2). The analysis showed that the best reference genes available under each treatment differed (Table 3). The top two reference genes in Control group were *TUA* and *GAPDH*, while *18S* ranked last. The two most stable genes were *EF1-α* and *CDC25* across the SA treatment and *18S* at the bottom, the least stable. In the MeJA treatment, *UBC* and *Actin* showed the highest ranking while *TUB* showed the lowest ranking. *Actin* and *PP2A* in the ETH treatment group were the most stable genes and the least stable gene was *EF1-α*. Across the ABA treatment group, *Actin* and *SAND* ranked the most stable, with *GAPDH* ranked last. Furthermore, from the Total group, *UBC*, *Actin*, *18S,* and *PP2A* were the top ranked genes, which correlated well with the results obtained from the four hormone treatments.

### 2.4. Validation of the Stability of Reference Genes

To validate the reliability of reference genes, the expression of *STR* was normalized using these genes (Figure 6). *STR*, a central gene encoding the enzyme that catalyzes the rate-limiting step of MIAs, can strongly influence the production of MIAs. Hormones like MeJA, which cause changes in the external environment, lead to changes in the expression of *STR* (Cao and Wang 2021). Since its expression is associated with changes in the MIA content, the accumulation of gelsemium alkaloids also undergoes relevant changes.

The relative expression patterns of *STR* showed similar trends when the two most stable genes were used individually or in combination as the reference genes for normalization. However, when the two unstable genes were used as references, the relative expression levels of *STR* showed significant fluctuations. For instance, under ETH treatment for 0–48 h, the expression level of *STR* was the highest at 0 h and the lowest at 8 h when using the stable genes (*Actin*, *PP2A* and *Actin* + *PP2A*) as the reference genes, with no significant change at 48 h compared to 0 h; this was shown to be the same as gelsenicine. In contrast, the expression level of *STR* was highest at 48 h and the general trends were different when using the least stable genes (*UBC* and *EF1-α*). In addition, the other three hormone treatment groups were validated on the basis of the results obtained in Table 3 for the most and least stable genes. However, the expression levels and trends of *STR* under different groups pointed to similar conclusions. It is evident that using unstable reference genes for gene expression analysis in *G. elegans* can lead to unreliable results.

### 2.5. Expression Patterns of Pathway Genes Involved in the Biosynthesis of Gelsenicine

To further enrich the study of the biosynthetic pathways involved in MIAs, 15 genes related to the upstream pathways were screened based on genomic and transcriptome datasets and used the “genome + transcriptome + metabolome” co-expression association analysis method (descriptions for the 15 genes are shown in Table 4). The expression patterns of related enzyme genes on the MIA biosynthetic pathway were analyzed using the two most stable genes from each treatment as references to identify potential links between gelsenicine and MIA biosynthesis pathways. After different hormone treatments, almost all gene expressions changed to a certain extent over the six time periods between 0–48h. The results are presented as a combination of heatmaps and correlation analyses based on the trend of relative expression (Figure 7, Figure 8 and Figure 9).

#### 2.5.1. Co-Expression Screening for MIA Pathway Genes

The co-expression association analysis is highly predictive for the screening of multi-gene or super-gene family candidates and allows for a more precise study of key sequences. Based on established research, the trend in the relative content of gelsenicine in different parts of *G. elegans* was highest in roots, followed by flowers, stems, and leaves in that order. We used *8-HGO* and gelsenicine for association analysis as an example and found ten *8-HGO*-related candidate sequences after identification of genomic data. From the results (Figure 10), there are three genetic sequences that are likely to be involved in gelsenicine biosynthesis. As the two nearest genes clustered with gelsenicine were lowly expressed in the leaf based on transcriptome information, the focus sequence Contig3.509, which is likely to be involved in gelsenicine biosynthesis, was selected. Other genes were screened by the same approach.

The expression patterns of genes coding for enzyme in the MIA biosynthesis pathway was analyzed by using the two most stable genes from each treatment as references to identify potential links between gelsenicine and MIA biosynthesis pathways. After different hormone treatments, almost all gene expressions were changed to a certain extent during the six time periods between 0–48 h. A combination of heatmaps and correlation analyses based on the trend of relative expression is shown (Figure 7, Figure 8 and Figure 9).

#### 2.5.2. Expression Pattern Analysis of Gelsenicine-Related Genes under SA Treatments

Among the SA treatments, the 15 genes with the most similar expression patterns to gelsenicine accumulation were *SGD*, *STR,* and *LAMT*, which all showed a general trend of decreasing–increasing–decreasing and were finally significantly down-regulated at 48 h compared to 0 h (Figure 7a). Clearly, *SGD* expression showed a quite significant positive correlation with gelsenicine accumulation, with both *STR* and *LAMT* expression being quite similar (Figure 8a and Figure 9a). Conversely, for instance, *AnPRT*, *8-HGO*, and other six genes nearly all showed significantly increasing expression at 2 h and 48 h (Figure 7a). These genes were not consistent with changes in gelsenicine and showed a significant negative correlation.

#### 2.5.3. Expression Pattern Analysis of Gelsenicine-Related Genes under MeJA Treatments

After MeJA treatment, the changes of gelsenicine content were mainly up-regulated significantly at 8 h and down-regulated significantly at 4 h and 48 h (Figure 2). The trends of *GES* and *AnPRT* were similar to that of gelsenicine, while the trends of *8-HGO*, *7-DLS*, and another four genes were also similar to that of gelsenicine, with a significant increase and peak at 2 h; most of them were significantly down-regulated at 48 h compared to 0 h (Figure 7b). Interestingly, there was a highly significant positive correlation between *GES*, *AnPRT,* and *8-HGO* with gelsenicine, as well as a significant positive correlation between *7-DLS*, *SLS*, and another four genes with gelsenicine (Figure 8b and Figure 9b). This suggests that with MeJA treatment, it is likely that all of these genes are positively connected with gelsenicine production, although there are differences in the degree of response.

#### 2.5.4. Expression Pattern Analysis of Gelsenicine-Related Genes under ETH Treatments

Following ETH hormone treatment, gelsenicine content increased significantly at 24 h, and finally there was no significant change at 48 h compared with 0 h (Figure 2). Most of the expression profiles showed a tendency to decrease and then increase in expression, such as *8-HGO*, *LAMT*, *STR,* and another four genes, which reached their lowest expression level at 2 h or 8 h (Figure 7c). The expression profiles of *AS*, *PRAI,* and *TSA* genes were very similar, which rebounded at 8 h after decreasing at 0 h, with the expression of most of them then significantly decreasing at 48 h compared with 0 h (Figure 7c). Their overall expression profile was positively correlated with gelsenicine (Figure 8c and Figure 9c). In addition, ETH treatment indicated up-regulation of the most MEP pathway-related genes versus down-regulation of the indole pathway-related genes, which may account for the lack of significant changes in gelsenicine content.

#### 2.5.5. Expression Pattern Analysis of Gelsenicine-Related Genes under ABA Treatments

Under ABA treatment, gelsenicine content decreased and then increased at 48 h and was up-regulated by 1.11-fold relative to 0 h (Figure 2). Similar to the gelsenicine change, *8-HGO*, *LAMT,* and *STR* gene expression profiles showed an overall trend of decreasing and then increasing (Figure 7d), in which *8-HGO* showed a highly significant positive correlation with gelsenicine, and the other two genes showed a significant positive correlation with gelsenicine (Figure 8d and Figure 9d). Conversely, *IS* and another three gene expression profiles first rose, peaking at 8 h and then decreased (Figure 7d). The trends of *TSB*, *IGPS,* and *SLS* were not exactly the same from those genes, as they showed a complex response (Figure 7d). Under the influence of ABA, the genes responded at varying degrees, but ultimately there was an up-regulation of most genes, which may have been responsible for the significant up-regulation of gelsenicine levels.

Overall, the expression patterns between gelsenicine and the genes on the pathway were somewhat correlated at different time intervals after different treatments. In particular, the expression patterns of *8-HGO*, *LAMT,* and *STR* had the most similar expression profile relative to gelsenicine content, suggesting that these genes may be involved in coordinating the biosynthetic process of gelsenicine, as well as potentially the biosynthetic pathways of other MIA classes in *G. elegans*.

## 3. Discussion

Gene expression analysis is a key technique to access the connection between the changes in plant secondary metabolism and the activity of related genes in the biosynthesis pathway [44]. RT-qPCR is one of the most accurate technique to analyze gene expression profiles, and the screening of optimal reference genes is a crucial step [45]. In many plant species, optimal reference genes have been identified under various experimental conditions [46]. At present, *EF-1α* and *TUA* are the best two genes for *Chenopodium quinoa* seedlings after treating with ABA [47]. *EF-1α,* and *TUB* were selected as the best reference genes for studying *Petroselinum crispum* gene expression profiles under different hormone treatments and abiotic stress [48]. *Actin* was found to be the most stable reference gene in *Celery* subjected to various abiotic stresses as well as hormone treatments [49]. Thus, reference genes can be different for different experimental conditions. In this work, we selected ten candidate reference genes to identify the most suitable reference genes in *G. elegans* under specific conditions, thus contributing to the gelsenicine biosynthesis pathway-related genes for expression pattern analysis by RT-qPCR. Our results showed that *EF1-α* and *CDC25*, *UBC* and *Actin*, *Actin* and *PP2A*, *Actin* and *SAND* were the optimal reference genes under SA, MeJA, ETH, and ABA, respectively. In contrast, *18S*, *TUB*, and *GAPDH* were considered to be inappropriate genes for normalization in *G. elegans* leaves under SA, MeJA, and ABA respectively. Notably, *EF1-α* showed the least stability in ETH treatment, which is opposite to the SA treatment. As identifying reference genes in different experimental conditions is crucial for the accuracy of RT-qPCR studies, our work lays the foundation for future gene expression analysis and functional studies of *G. elegans*.

To ensure the reliability and accuracy of the results of reference gene stability analysis, GeNorm, NormFinder, BestKeeper, ∆CT, and RefFinder tools are commonly used. However, the genes identified as stable depend on the specific analysis performed. For example, under SA treatment, analysis by NormFinder and ∆CT methods identified that the most stable genes were *EF1-α* and *GAPDH*, GeNorm placed these two genes in second and fifth place, while BestKeeper placed these genes in third and first place. Although there were slight differences between rankings of these genes assessed by the various algorithms, the top five stable genes across these algorithms were similar for each group. For instance, *UBC*, *TUA*, *Actin*, *GAPDH,* and *EF1-α* were the top five stable genes based on the BestKeeper and ∆CT under the MeJA treatment, and NormFinder showed similar results except for *EF1-α*. In addition to *GAPDH*, the other four genes are also the most stable in GeNorm. A number of other studies showed similar differences between the results of these tools that occur from the use of different statistical approaches [50,51]. We found that RefFinder results were ranked similar to these four tools, which further supports the reliability and accuracy of these results. It is noteworthy that the V value obtained in GeNorm played an important role in deciding the optimal number for selecting the reference gene. In this study, the optimal number of reference genes selected for the four treatments was two genes, and the optimal reference gene combinations were the top two genes according to RefFinder (Table 3). A further validation using *STR* indicated that we had identified the best reference gene combinations for use in *G. elegans* leaves under the different hormone treatments that we studied.

As gelsenicine is formed from the MIA upstream biosynthesis pathway via a series of reactions, we carried out co-expression analysis of the “genome + transcriptome + metabolome” dataset to find candidate upstream pathway-related genes, whose expression profiles could be associated with the content accumulation of gelsenicine in different tissues of *G. elegans*. Previously, some studies had used similar co-expression analysis to identify pathway-associated genes. In *Salvia miltiorrhiza*, it was hypothesized that tanshinone biosynthetic enzymes would be expressed in the rhizome, due to the accumulation of tanshinones in this organ, and six of the probable 14 *CYP450s* were then further screened on this basis. One *CYP450* could finally be validated as being related to tanashinone biosynthesis [52]. In another study in *Siraitia grosvenorii*, using the terms “genome + transcriptome + metabolome” seven candidate *CYP450s* (out of 80) and five candidate *UDPGs* (out of 90) with potential to be involved in the mogroside V biosynthesis pathway were targeted for their successful association with this pathway [53,54,55]. In this work, we used co-expression analysis to identify 15 candidate genes related to the gelsenicine biosynthesis pathway. We found that the expression levels of these genes as well as the gelsenicine content significantly changed under different hormone treatments. We hypothesize that candidate gelsenicine biosynthesis related genes that are strongly associated with changes in content are more likely to be involved in the gelsenicine biosynthesis. *8-HGO*, *LAMT,* and *STR* expression profiles were similar to the variation trends in the content of gelsenicine, suggesting these genes are likely to be involved in coordinating the biosynthetic process of gelsenicine. For instance, after treating with ABA, the expression pattern of *8-HGO*, *LAMT,* and *STR* showed good correlation with the content trend of gelsenicine. These genes will be subjected to further molecular examination to determine their actual association with the gelsenicine biosynthesis. If any one of these three candidates is important for gelsenicine biosynthesis, this methodology with be of great use for quick identification of other MIA associated genes in *G. elegans*.

## 4. Materials and Methods

### 4.1. Plant Materials

Leaves from healthy *G. elegans* seedlings were collected from the experimental forestry farm of Hunan Academy of Forestry (113°06′ E, 28°12′ N) in Changsha, Hunan Province, China. Starting on 15 February 2023, 6-month-old seedlings were treated with SA, MeJA, ETH, and ABA at a concentration of 100 μM [36,37,56]. The material was sampled sequentially at 0 h (8 a.m.), 2 h (10 a.m.), 4 h (12 p.m.), 8 h (16 p.m.), 24 h (8 a.m.) and 48 h (8 a.m.). All samples were immediately frozen in liquid nitrogen and later stored at −80 °C for further sampling and analysis. A minimum of three independent biological replicates was obtained for all sampling sites. 

### 4.2. Determination of the Content of Gelsenicine by HPLC

Samples were fully ground into powder under liquid nitrogen and dried in a MODULYOD freeze dryer (Thermo Scientific, Santa Clara, CA, USA). Samples to be analyzed were weighed (2 g) and extracted twice by ultrasonication with 80% ethanol (1:25, *w*/*v*) for 0.5 h at 60 °C. The extraction solutions were combined for filtration, using 1mL as the sample solution. The prepared MeOH (stock) solution with 1 mg gelsenicine mL^−1^ (Chengdu Must Bio-technology Co., Chengdu, China) was added to give a final concentration for treatments of 0.04 mg·mL^−1^.

Analysis was performed on a Thermo Scientific Ultimate 3000 HPLC instrument (Thermo Scientific, Santa Clara, CA, USA). Samples were separated on an Agilent C_18_ column (5 μm, 4.6 mm × 250 mm). The flow rate was 1 mL·min^−1^, and the column temperature was maintained at 30 °C. The mobile phase consisted of acetonitrile (B) and phosphoric acid aqueous solution (D). The gradient elution for 40 min was as follows: 0–2 min, 10% B; 2–7 min, 10% B to 15% B; 7–30 min, 15% B; 30–40 min, 15% B to 90% B; 40–43 min, 90% B; 43.01–50 min, 10% B. The injection volume was 5 μL. Five concentrations of gelsenicine were injected, and then calibration curves were set by plotting the peak area versus the concentration. The sample solution was passed through a 0.22 μM organic microporous membrane filter (Tianjin Jinteng Experimental Equipment Co., Tianjin, China), and was analyzed at the above chromatographic conditions. Combined with the HPLC results and the linear relationship, the content of gelsenicine under different treatments was calculated. All samples were biologically repeated three times.

### 4.3. RNA Extraction and cDNA Synthesis

Total RNA was extracted from samples of *G. elegans* by the *SteadyPure* Plant RNA Extraction Kit (Hunan Acres Biological Engineering Co., Changsha, China). RNA mass and RNA purity were measured in an ultra-micro spectrophotometer (item no. 912A0959, purchased from BIO-DL, Shanghai, China) and RNA integrity was assessed by electrophoresis on a 1.0% agarose gel. cDNA from samples of *G. elegans* was synthesized according to the *Evo M-MLV* RT Mix Kit with the gDNA Clean product protocol (Hunan Acres Bioengineering Co., Changsha, China) and the amount of total RNA used was 700 ng.

### 4.4. Candidate Gene Selection and Primer Design

Based on our existing genomic database, we screened ten candidate reference genes, including *18S*, *GAPDH*, *Actin*, *TUA*, *TUB*, *SAND*, *EF1-α*, *UBC*, *PP2A,* and *CDC25* (Table 1). Fifteen target genes related to the upstream biosynthetic pathway of MIAs, namely *GES*, *G8H*, *8-HGO*, *ISY*, *7-DLS*, *LAMT*, *SLS*, *AS*, *AnPRT*, *PRAI*, *IGPS*, *TSA*, *TSB*, *STR,* and *SGD*, were also screened by a “genome + transcriptome + metabolome” co-association analysis (Table 4). Primers for RT-qPCR were designed by Beacon Designer 7 (Table 1 and Table 4) and synthesized at Shanghai Sangon Biotech Co., Ltd. (Shanghai, China). The primers had melting temperatures between 53–60 °C with a GC content spanning 40–55%. All efficiencies had to be at the range of 90–120%. The coding sequences of the 10 reference genes, and 15 target genes were provided by the genomics database (Bioproject ID PRJNA505365). Melting curves for all genes were obtained by RT-qPCR (Appendix A). 

### 4.5. RT-qPCR

RT-qPCR was performed with the ABI 7300 Real-Time Fluorescence PCR System (Applied Biosystems, Foster City, CA, USA) using the SYBR^®^ Green Premix *Pro Taq* HS qPCR Kit (Hunan Acres Biological Engineering Co., Changsha, China). A total of 20 μL of reaction mixture containing 2 μL of diluted cDNA (~25 ng/μL), 10 μL of 2 × SYBR Green *Pro Taq* HS premix, 0.4 μL of forward primer (10 μM), 0.4 μL of reverse primer (10 μM), 0.4 μL of ROX (20 μM), and 6.8 μL of ddH_2_O was used. PCR reaction conditions were as follows: 95 °C for 3 min, 40 cycles at 95 °C for 30 s, and 60 °C for 30 s. Melting curves were generated by heating the amplicon from 60 °C to 95 °C to confirm primer specificity. Each RT-qPCR reaction was repeated with three technical replicates and Ct was measured automatically. Relative changes in gene expression were calculated using the method of comparing 2^−ΔΔCT^ [57]. Additionally, a 5-fold dilution series (5^0^, 5^−1^, 5^−2^, 5^−3^ and 5^−4^) of template cDNA was used for all samples and (~100 ng/μL) for the standard curve. Correlation coefficients (*R^2^*) were measured from the standard curve based on these cDNA templates. The primer specificity was verified by the presence of a single peak in the melting curve analysis. Amplification efficiency (E) of the primer pair could be calculated from the slope value with the following formula: E = 10^−1÷*k*^.

### 4.6. Evaluation of Reference Genes

When performing GeNorm and NormFinder analyses, the original Ct value is first converted into a relative quantitative value (Q value) with the following equation Q = 2^−∆CT^ [58,59]. The expression stability (M) value and the pairwise variation (V) value are used for GeNorm. A smaller M value reflecting higher stability, and V value (V_*n*/*n*+1_ < 0.15 is a cut-off) determines the optimum number of reference genes to select. NormFinder also obtains the M values based on the variation of reference genes within and between samples [60]. BestKeeper requires the raw Ct value, which is then assessed by the standard deviation (SD) and coefficient of variance (CV) as the criteria for stable expression of the reference gene [61]. The ΔCT method assesses gene expression stability by calculating the mean standard deviation (SD). The RefFinder website (http://blooge.cn/RefFinder/, accessed on 30 November 2022) combined the above four algorithms to select the most suitable reference genes [62]. 

### 4.7. Validation of Reference Genes

To verify the stability of the screened reference genes, *STR* was selected as the target gene and the relative expression levels of *STR* under different treatments were observed by using the two most stable and the least stable reference genes. The expression patterns of 15 genes related to the upstream biosynthesis pathway of MIAs were also further analyzed under different treatments. The Ct values obtained from the raw data were also calculated based on 2^−ΔΔCT^. Three biological replicates were performed for each sample. In addition, relevant graphs were produced by using Prism9.0.0 software, TBtools2.012, Adobe Illustrator 2021, and OmicStudio online website (https://www.omicstudio.cn/, accessed on 8 June 2023), while statistical analysis was carried out using SPSS 17.0 software.

## 5. Conclusions

The identification of functional reference genes is a prerequisite for qualification of gene expression in specific experimental conditions when using RT-qPCR. In this research, we calculated the gene expression stability of ten candidate reference genes of *G. elegans* leaves with or without the application of one of four hormone treatments associated with changes in gelsenicine biosynthesis. Our data recommends that two reference genes be used to normalize RT-qPCR data. When leaves were treated with SA, *EF1-α* and *CDC25* should be used as reference genes whereas, *UBC* and *Actin* were the best reference genes for the MeJA treated leaves, and *Actin* and *PP2A* for ETH treated samples. *Actin* and *SAND* were identified to be the most suitable genes when *G. elegans* leaves were treated with ABA. Our research helps to accurately quantify the biosynthesis pathway-related genes of MIAs in *G. elegans* and provides a basis for further analyses by RT-qPCR of genes that participate in the complex regulation of the MIA biosynthesis pathway.

## Figures and Tables

**Figure 1 ijms-24-15973-f001:**
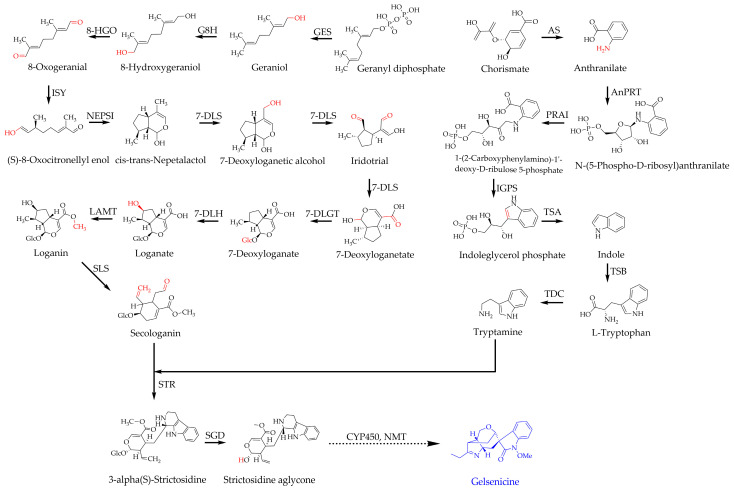
The biosynthesis pathway of MIAs in *G. elegans* (solid arrows represent known synthetic pathways and the dashed arrows represent speculative pathways). GES: Geraniol synthase; G8H: Geraniol 8-hydroxylase; 8-HGO: 8-hydroxygeraniol dehydrogenase; ISY: (S)-8-oxocitronellyl enol synthase; NEPSI: (+)-cis,trans-nepetalactol synthase; 7-DLS: 7-deoxyloganetic acid synthase; 7-DLGT: 7-deoxyloganetic acid glucosyltransferase; 7-DLH: 7-deoxyloganate 7-hydroxylase; LAMT: Loganate methyltransferase; SLS: Secologanin synthase; AS: Anthranilate synthase; AnPRT: Anthranilate phosphoribosyltransferase; PRAI: Phosphoribosylanthranilate isomerase; IGPS: Indole-3-glycerol phosphate synthase; TSA: Tryptophan synthase alpha chain; TSB: Tryptophan synthase beta chain; TDC: L-tryptophan decarboxylase; STR: Strictosidine synthase; SGD: Strictosidine-β-D- glucosidase; CYP450: Cytochrome P450 enzymes; NMT: N-methyltransferase.

**Figure 2 ijms-24-15973-f002:**
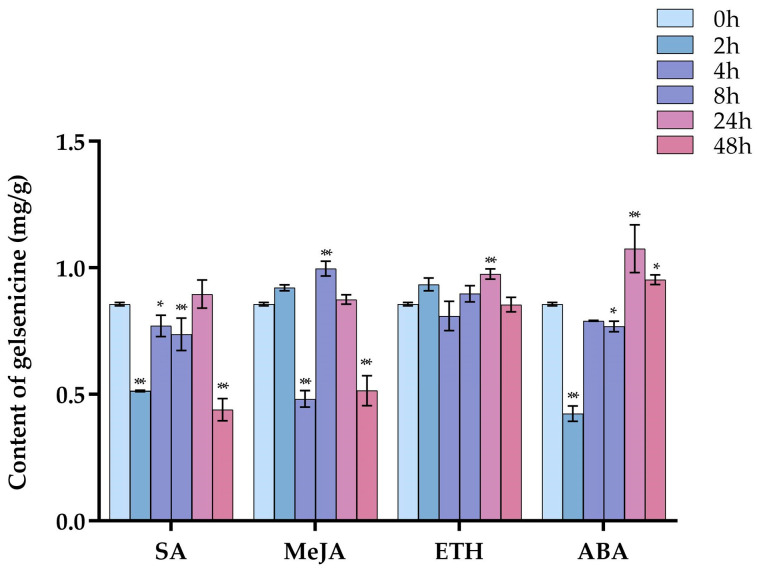
The content of gelsenicine under four hormone treatments at 0–48 h. The (*) indicates *p* < 0.05, and the (**) indicates *p* < 0.01.

**Figure 3 ijms-24-15973-f003:**
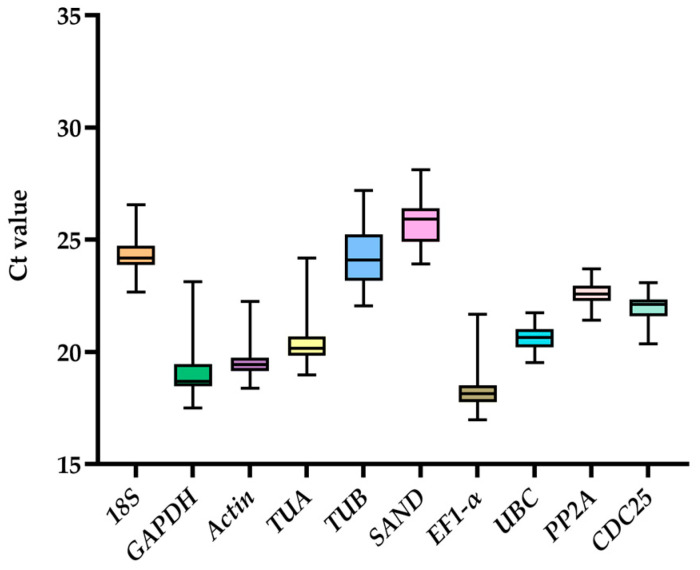
Distribution of Ct values among the 10 candidate reference genes. Lines shown in the box-plot graph of Ct value display the median values. Lower and upper boxes represent the 25th percentile to the 75th percentile. Whiskers indicate the maximum and minimum values.

**Figure 4 ijms-24-15973-f004:**
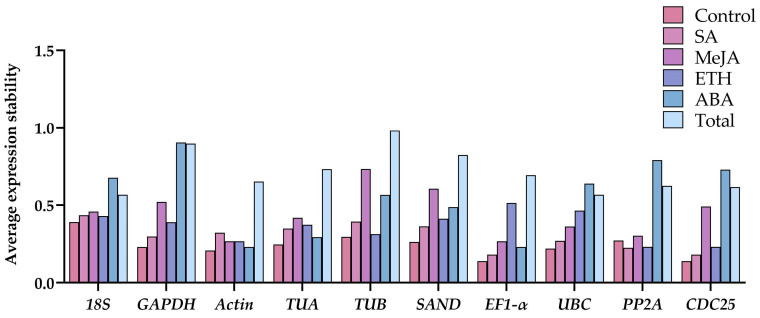
Average expression stability values (M) of 10 candidate reference genes using GeNorm.

**Figure 5 ijms-24-15973-f005:**
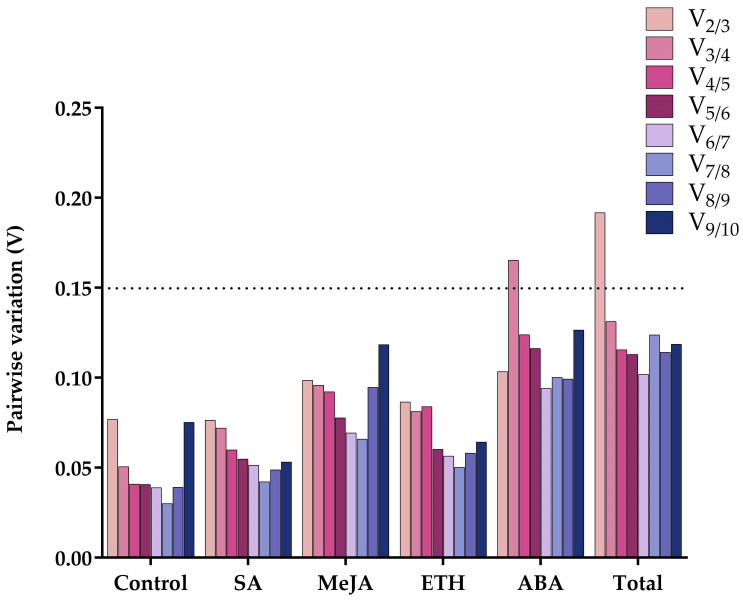
Pairwise variation (V) values obtained by GeNorm analysis on 10 candidate reference genes in the six groups.

**Figure 6 ijms-24-15973-f006:**
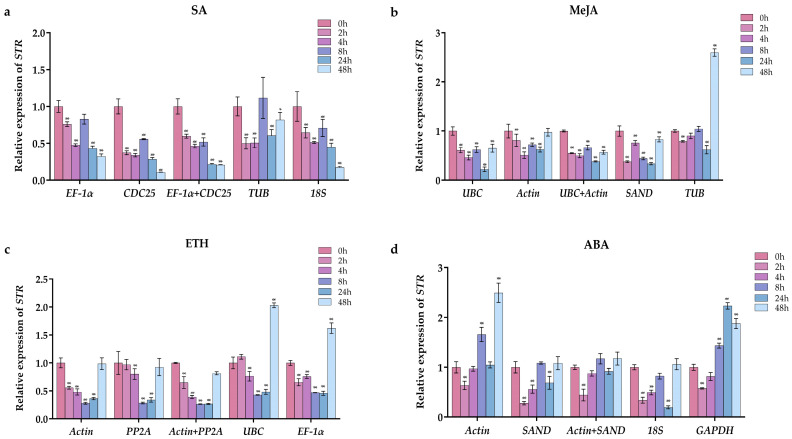
Validation of the reference gene by the relative expression of the target gene *STR* in different hormone treatments. Graphs (**a**–**d**) are the results of SA, MeJA, ETH and ABA. The (*) indicates *p* < 0.05, and the (**) indicates *p* < 0.01.

**Figure 7 ijms-24-15973-f007:**
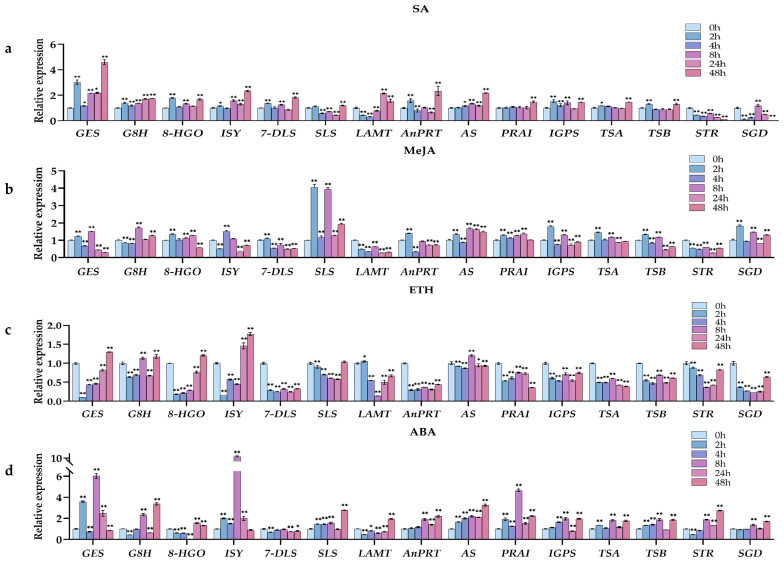
(**a**) The results for the expression patterns of 15 candidate genes related to the upstream pathway of MIA biosynthesis at 0–48 h of SA treatment; (**b**) the results for the expression patterns of 15 candidate genes related to the upstream pathway of MIA biosynthesis at 0–48 h of MeJA treatment; (**c**) the results for the expression patterns of 15 candidate genes related to the upstream pathway of MIA biosynthesis at 0–48 h of ETH treatment; (**d**) the results for the expression patterns of 15 candidate genes related to the upstream pathway of MIA biosynthesis at 0–48 h of ABA treatment. The (*) indicates *p* < 0.05, and the (**) indicates *p* < 0.01.

**Figure 8 ijms-24-15973-f008:**
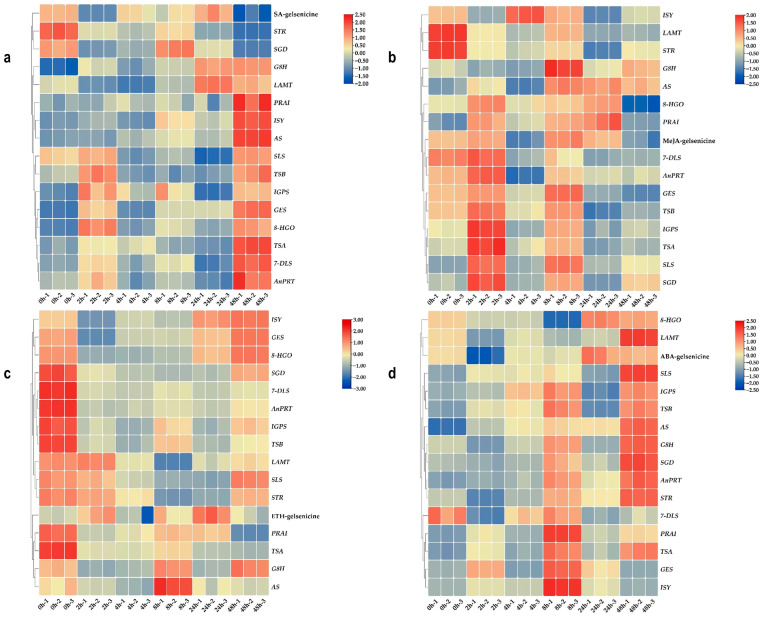
(**a**) The results for the heatmaps of gelsenicine and the 15 upstream biosynthetic pathway genes associated with it under SA treatment; (**b**) the results for the heatmaps of gelsenicine and the 15 upstream biosynthetic pathway genes associated with it under MeJA treatment; (**c**) the results for the heatmaps of gelsenicine and the 15 upstream biosynthetic pathway genes associated with it under ETH treatment; (**d**) the results for the heatmaps of gelsenicine and the 15 upstream biosynthetic pathway genes associated with it under ABA treatment. The darker the colors, the stronger is the correlation.

**Figure 9 ijms-24-15973-f009:**
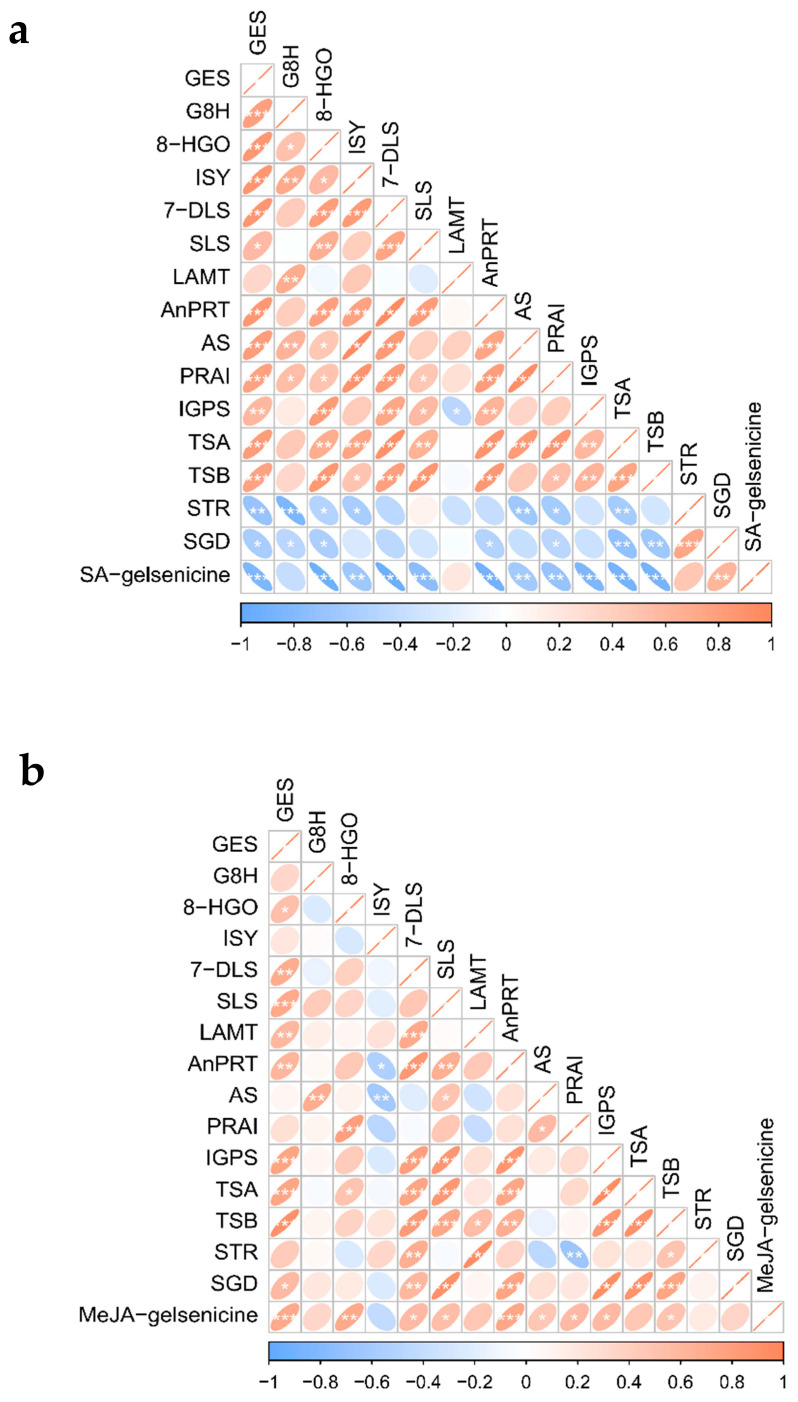
(**a**) The results for the correlation analyses between gelsenicine and 15 upstream biosynthetic pathway genes under SA treatment; (**b**) the results for the correlation analyses between gelsenicine and 15 upstream biosynthetic pathway genes under MeJA treatment; (**c**) the results for the correlation analyses between gelsenicine and 15 upstream biosynthetic pathway genes under ETH treatment; (**d**) the results for the correlation analyses between gelsenicine and 15 upstream biosynthetic pathway genes under ABA treatment. Thick lines (weak correlation), thin lines (strong correlation), red and rightward slash (positive correlation), blue and leftward slash (negative correlation), only the more relevance cases are shown with asterisks [43]. The darker the colors, the more asterisks, the stronger is the correlation. The (*) indicates *p* < 0.05, and the (**) indicates *p* < 0.01. The (***) indicates *p* < 0.001.

**Figure 10 ijms-24-15973-f010:**
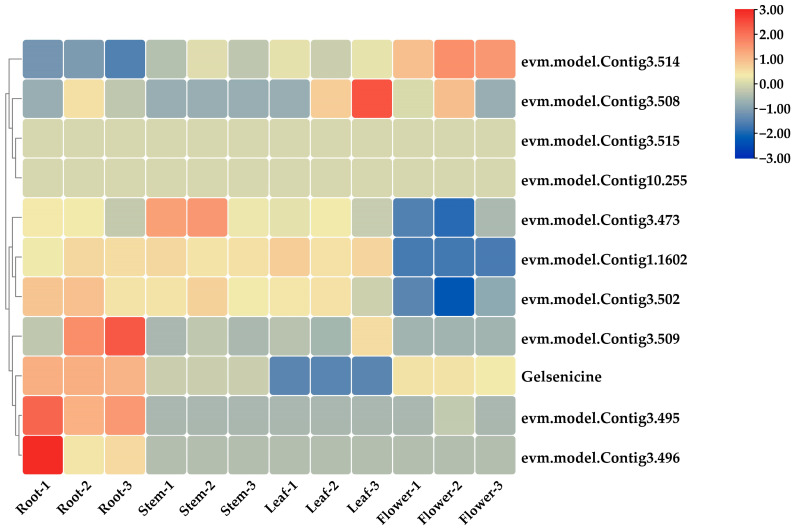
Screening of genes related to the biosynthetic pathway of MIAs based on co-expression correlation analysis, shown here with *8-HGO* as an example.

**Table 1 ijms-24-15973-t001:** Description and primer information for 10 candidate reference genes.

Gene	Gene Description	Primer Sequence	Slope/*k*	E/%	*R* ^2^	Accession Number
*18S*	18S Ribosomal RNA	F: GATGGAGTCCCGAAGTTGC R: TCCAGATCGCATGGCATAG	−3.37	97.9	0.994	OR413515
*GAPDH*	Glyceraldehyde-3-phosphate dehydrogenase	F: AAGGGTGGTGCCAAGAAGG R: CAGTGGGAACACGGAAAGC	−3.11	109.8	0.998	OR413520
*Actin*	Actin	F: GTTGCCCAGAAGTCCTATT R: TTCCTGTGGACGATTGATG	−3.12	109.3	0.997	OR413517
*TUA*	α-Tubulin	F: ATGAAGTTAGAACAGGGACA R: CAAGCAGGGAGTGAGTAGA	−3.15	107.6	0.992	OR413519
*TUB*	β-Tubulin	F: TGTCCGTAAAGAAGCCGAGAA R: CAGGGAAACGAAGGCAACA	−3.26	102.8	0.996	OR413518
*SAND*	SAND family protein	F: CATCCGACCCACCTACCGT R: ACTCTGCCAACTCCGCTCC	−3.13	108.5	0.996	OR413523
*EF1-α*	Elongation factor 1α	F: AAGCCACTCCGTCTCCCACT R: TCGGCAAACTTGACAGCAATA	−3.1	110.1	0.991	OR413516
*UBC*	Ubiquitin C	F: CAAAGGTGGTGAGGAGGAT R: ACAGAGCAGCGACTGAATG	−3.33	99.6	0.994	OR413521
*PP2A*	Protein phosphatases 2A	F: TGATTACCTGCCTCTGAC R: TGTGGAACCTCCTGTATG	−3.19	105.9	0.999	OR413514
*CDC25*	Cell division cyclin 25 homolog C	F: CAGGGATGACGAAAGGAGT R: CGCAATGGAAAACAAGAGT	−3.11	109.8	0.999	OR413522

**Table 2 ijms-24-15973-t002:** Expression stability analysis of reference genes assayed by GeNorm, NormFinder, BestKeeper, ΔCT, and RefFinder. Stab.: gene stability value.

Group	Rank	GeNorm		NormFinder		BestKeeper		ΔCT		RefFinder	
		Gene	Stab.	Gene	Stab.	Gene	CV ± SD	Gene	Stab.	Gene	Stab.
Control	1	*CDC25*	0.139	*TUA*	0.016	*Actin*	0.41 ± 0.08	*TUA*	0.30	*TUA*	2.55
	2	*EF1-α*	0.139	*GAPDH*	0.126	*UBC*	0.42 ± 0.09	*GAPDH*	0.32	*GAPDH*	2.78
	3	*Actin*	0.207	*SAND*	0.147	*GAPDH*	0.73 ± 0.13	*UBC*	0.32	*Actin*	2.94
	4	*UBC*	0.220	*UBC*	0.176	*CDC25*	0.68 ± 0.15	*SAND*	0.33	*UBC*	3.13
	5	*GAPDH*	0.230	*Actin*	0.232	*PP2A*	0.69 ± 0.16	*Actin*	0.35	*CDC25*	3.72
	6	*TUA*	0.246	*PP2A*	0.263	*EF1A*	0.90 ± 0.16	*CDC25*	0.37	*EF1-α*	4.56
	7	*SAND*	0.263	*TUB*	0.266	*TUA*	0.92 ± 0.19	*PP2A*	0.38	*SAND*	5.09
	8	*PP2A*	0.272	*CDC25*	0.291	*SAND*	0.80 ± 0.20	*EF1-α*	0.38	*PP2A*	6.40
	9	*TUB*	0.296	*EF1-α*	0.293	*TUB*	1.42 ± 0.33	*TUB*	0.40	*TUB*	8.45
	10	*18S*	0.391	*18S*	0.749	*18S*	2.82 ± 0.70	*18S*	0.77	*18S*	10.00
SA	1	*CDC25*	0.181	*EF1-α*	0.164	*GAPDH*	0.90 ± 0.17	*EF1-α*	0.35	*EF1-α*	1.32
	2	*EF1-α*	0.181	*GAPDH*	0.191	*CDC25*	0.77 ± 0.17	*GAPDH*	0.37	*CDC25*	2.06
	3	*PP2A*	0.225	*CDC25*	0.236	*EF1-α*	1.04 ± 0.19	*CDC25*	0.38	*GAPDH*	2.11
	4	*UBC*	0.270	*UBC*	0.254	*Actin*	1.20 ± 0.24	*UBC*	0.40	*PP2A*	4.40
	5	*GAPDH*	0.298	*PP2A*	0.286	*PP2A*	1.16 ± 0.26	*PP2A*	0.42	*UBC*	4.43
	6	*Actin*	0.322	*Actin*	0.294	*UBC*	1.65 ± 0.34	*Actin*	0.42	*Actin*	5.42
	7	*TUA*	0.349	*TUA*	0.326	*TUA*	1.85 ± 0.38	*TUA*	0.44	*TUA*	7.00
	8	*SAND*	0.363	*SAND*	0.338	*SAND*	1.56 ± 0.40	*SAND*	0.45	*SAND*	8.00
	9	*TUB*	0.394	*TUB*	0.411	*TUB*	1.72 ± 0.41	*TUB*	0.51	*TUB*	9.00
	10	*18S*	0.435	*18S*	0.524	*18S*	1.93 ± 0.49	*18S*	0.60	*18S*	10.00
MeJA	1	*Actin*	0.267	*UBC*	0.174	*UBC*	1.08 ± 0.22	*UBC*	0.56	*UBC*	1.41
	2	*EF1-α*	0.267	*TUA*	0.267	*GAPDH*	1.61 ± 0.30	*TUA*	0.61	*Actin*	2.63
	3	*PP2A*	0.303	*GAPDH*	0.292	*TUA*	1.58 ± 0.32	*Actin*	0.61	*TUA*	2.78
	4	*UBC*	0.362	*Actin*	0.360	*Actin*	1.73 ± 0.33	*GAPDH*	0.64	*EF1-α*	3.50
	5	*TUA*	0.419	*18S*	0.418	*EF1-α*	2.06 ± 0.37	*EF1-α*	0.64	*GAPDH*	3.72
	6	*18S*	0.458	*EF1-α*	0.420	*PP2A*	1.65 ± 0.37	*PP2A*	0.66	*PP2A*	5.24
	7	*CDC25*	0.491	*PP2A*	0.459	*18S*	1.55 ± 0.37	*18S*	0.68	*18S*	6.19
	8	*GAPDH*	0.521	*CDC25*	0.657	*CDC25*	2.37 ± 0.51	*CDC25*	0.79	*CDC25*	7.74
	9	*SAND*	0.606	*SAND*	0.738	*SAND*	2.08 ± 0.55	*SAND*	0.90	*SAND*	9.00
	10	*TUB*	0.733	*TUB*	1.175	*TUB*	3.54 ± 0.90	*TUB*	1.24	*TUB*	10.00
ETH	1	*CDC25*	0.231	*Actin*	0.183	*Actin*	0.68 ± 0.13	*Actin*	0.42	*Actin*	1.32
	2	*PP2A*	0.231	*PP2A*	0.208	*PP2A*	1.01 ± 0.23	*PP2A*	0.43	*PP2A*	1.68
	3	*Actin*	0.267	*CDC25*	0.269	*CDC25*	1.19 ± 0.26	*CDC25*	0.45	*CDC25*	2.28
	4	*TUB*	0.313	*TUB*	0.322	*18S*	1.34 ± 0.33	*TUB*	0.48	*TUB*	4.43
	5	*TUA*	0.373	*TUA*	0.330	*GAPDH*	1.80 ± 0.33	*TUA*	0.49	*TUA*	5.62
	6	*GAPDH*	0.390	*GAPDH*	0.340	*TUB*	1.52 ± 0.35	*GAPDH*	0.49	*GAPDH*	5.73
	7	*SAND*	0.412	*SAND*	0.347	*SAND*	1.46 ± 0.36	*SAND*	0.51	*18S*	6.73
	8	*18S*	0.430	*18S*	0.408	*TUA*	1.89 ± 0.38	*18S*	0.54	*SAND*	7.00
	9	*UBC*	0.465	*UBC*	0.521	*EF1-α*	2.13 ± 0.39	*UBC*	0.62	*UBC*	9.24
	10	*EF1-α*	0.514	*EF1-α*	0.633	*UBC*	2.77 ± 0.56	*EF1-α*	0.71	*EF1-α*	9.74
ABA	1	*Actin*	0.230	*SAND*	0.313	*PP2A*	1.72 ± 0.40	*Actin*	0.73	*Actin*	2.14
	2	*EF1-α*	0.230	*TUB*	0.337	*CDC25*	1.87 ± 0.42	*SAND*	0.76	*SAND*	2.51
	3	*TUA*	0.294	*Actin*	0.353	*UBC*	2.49 ± 0.52	*TUB*	0.77	*TUB*	3.31
	4	*SAND*	0.487	*UBC*	0.480	*TUB*	2.47 ± 0.61	*UBC*	0.79	*EF1-α*	3.76
	5	*TUB*	0.566	*EF1-α*	0.540	*SAND*	2.84 ± 0.75	*EF1-α*	0.81	*UBC*	4.12
	6	*UBC*	0.639	*18S*	0.631	*18S*	3.34 ± 0.81	*18S*	0.88	*PP2A*	5.20
	7	*18S*	0.677	*TUA*	0.701	*Actin*	4.50 ± 0.92	*TUA*	0.91	*CDC25*	5.66
	8	*CDC25*	0.729	*CDC25*	0.764	*EF1-α*	5.49 ± 1.06	*CDC25*	0.96	*TUA*	6.03
	9	*PP2A*	0.790	*PP2A*	0.891	*TUA*	5.64 ± 1.22	*PP2A*	1.07	*18S*	6.24
	10	*GAPDH*	0.904	*GAPDH*	1.251	*GAPDH*	6.03 ± 1.26	*GAPDH*	1.36	*GAPDH*	10.00
Total	1	*18S*	0.567	*Actin*	0.420	*PP2A*	1.77 ± 0.40	*Actin*	0.81	*UBC*	1.68
	2	*UBC*	0.567	*UBC*	0.453	*UBC*	2.16 ± 0.45	*UBC*	0.85	*Actin*	2.24
	3	*CDC25*	0.616	*TUA*	0.554	*CDC25*	2.07 ± 0.45	*TUA*	0.90	*18S*	2.83
	4	*PP2A*	0.624	*18S*	0.601	*18S*	2.02 ± 0.49	*18S*	0.91	*PP2A*	3.16
	5	*Actin*	0.652	*PP2A*	0.606	*Actin*	2.93 ± 0.57	*PP2A*	0.91	*TUA*	4.58
	6	*EF1-α*	0.693	*EF1-α*	0.612	*EF1-α*	3.70 ± 0.68	*EF1-α*	0.91	*CDC25*	4.58
	7	*TUA*	0.732	*CDC25*	0.715	*TUA*	3.82 ± 0.78	*CDC25*	0.96	*EF1-α*	6.00
	8	*SAND*	0.823	*SAND*	0.792	*SAND*	3.43 ± 0.88	*SAND*	1.06	*SAND*	8.00
	9	*GAPDH*	0.897	*GAPDH*	0.989	*TUB*	4.33 ± 1.05	*GAPDH*	1.19	*GAPDH*	9.24
	10	*TUB*	0.982	*TUB*	1.167	*GAPDH*	5.46 ± 1.05	*TUB*	1.32	*TUB*	9.74

**Table 3 ijms-24-15973-t003:** Most stable and least stable combination of reference genes based on RefFinder.

Experimental Treatments											
Control		SA		MeJA		ETH		ABA		Total	
Most	Least	Most	Least	Most	Least	Most	Least	Most	Least	Most	Least
*TUA* *GAPDH*	*18S*	*EF1-α* *CDC25*	*18S*	*UBC* *Actin*	*TUB*	*Actin* *PP2A*	*EF1-α*	*Actin* *SAND*	*GAPDH*	*UBC* *Actin* *18S*	*TUB*

**Table 4 ijms-24-15973-t004:** Description and primer information for 15 candidate pathway genes.

Gene	Gene Description	Primer Sequence	Slope/*k*	E/%	*R* ^2^	Accession Number
*GES*	Geraniol synthase	F: GGCTGCGTTTCAGGTTGCT R: CTTTAGGTGGGCTTGGGTG	−3.20	105.2	0.983	OR413524
*G8H*	Geraniol 8-hydroxylase	F: GTTTGGCGGAACAGACACC R: CGCTGAAATCCCACTTGCT	−3.28	102.0	0.995	OR413525
*8-HGO*	8-hydroxygeraniol dehydrogenase	F: CTGTCTTCCCGCTGCTTGC R: CGTTCCATTGCCGTGTTGA	−3.24	103.5	0.993	OR413526
*ISY*	(S)-8-oxocitronellyl enol synthase	F: CCGACCTGCTCTGGTTTTC R: AGGCTCACCTGTTCTTTGC	−3.01	114.8	0.992	OR413527
*7-DLS*	7-deoxyloganetic acid synthase	F: TGGCTGAGGTGTTGTTTG R: TGAATACCAGGCGAGTTT	−3.24	103.3	0.995	OR413528
*LAMT*	Loganate methyltransferase	F: CTGCTCCACAGGTCCCAATA R: GTGCCATCAACCCTCCGT	−3.43	95.7	0.996	OR413529
*SLS*	Secologanin synthase	F: TAGGCTGCTATTTGGGGATT R: ATGAGCACGGCAGGTTTT	−3.38	97.7	0.995	OR413530
*AS*	Anthranilate synthase	F: GGCGAATCCCGTTGTTGT R: TTGAGGCGTTCCAGGTCC	−3.36	98.5	0.996	OR413531
*AnPRT*	Anthranilate phosphoribosyltransferase	F: ACGGCAATCCTCCTTCCAA R: TTCGCCTGAGCATCCAACA	−3.21	105.0	0.999	OR413532
*PRAI*	Phosphoribosylanthranilate isomerase	F: GGGGCTTGGCTATTCTTGT R: GTTTTCCTCAGAGCAGCGT	−3.39	97.3	0.993	OR413533
*IGPS*	Indole-3-glycerol phosphate synthase	F: TGGTCCCTTTGAGTTTCGG R: AGGCAACCCAGTTCGTGAG	−3.21	105.0	0.998	OR413534
*TSA*	Tryptophan synthase alpha chain	F: CAAGCGTGGTGTTGAAAAG R: GCTCTGGGAGTTGTGGGTG	−3.25	103.2	0.998	OR413535
*TSB*	Tryptophan synthase beta chain	F: CAGTTCATTCTGGGACCGC R: TTCCAATGCCTGCTTCCTT	−3.21	104.8	0.998	OR413536
*STR*	Strictosidine synthase	F: AAGGAAGAGGGCGTGGAA R: GCAACAGGCAATGCAGAA	−3.33	99.8	0.987	OR413537
*SGD*	Strictosidine-β-D- glucosidase	F: TATGGTTATGCGTCGGGTGT R: AAGGCTCTGTGCCAGGGTT	−3.39	97.1	0.996	OR413538

## Data Availability

The RNA-Seq data (Biosample ID from SAMN11089884 to SAMN11089892) were deposited in the National Center for Biotechnology Information (https://www.ncbi.nlm.nih.gov/bioproject/?linkname=biosample_bioproject&from_uid=11089884, accessed on 13 November 2018).

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
