# Peer review of "Reference Genes Screening and Gene Expression Patterns Analysis Involved in Gelsenicine Biosynthesis under Different Hormone Treatments in Gelsemium elegans"

_ijms, 2023, doi:10.3390/ijms242115973_

Round 1
Reviewer 1 Report
Comments and Suggestions for Authors
The research topic is actual and interesting. The study is focused on the validation of reference genes and on the co-expression patterns of gelsenicine biosynthetic genes after elicitation.
I would suggest the authors to shorten the manuscript title.
The introduction should be reduced to half. I suggest to focus on the state-of-art, emphasize the importance and novelty of the authors' results, and reduce the general statements.
Please consider the correctness of "genome + transcriptome + metabolome". Such a collocation evokes transcriptome and metabolic profiling, not a targeted study.
The objective should be clearly formulated at the end of Introduction.
Please consider the position of subsection 2.1. Please mention the source of gelsenicine standard.
It is not clear how the candidate biosynthetic genes were identified in G. elegans genome. Were the sequences predicted by homology-based search? Please specify the dataset containing genomic data (database, acc. number). Similarly, the selected contigs - were these sequences functionally annotated?
Author Response
Dear reviewers,
Thank you for your useful suggestion. We have revised it carefully according to your suggestions.Please see the attachment.
Your suggestion of our manuscript will be deeply appreciated, and we look forward to receiving reply from you.
Thank you and best regards.
PhD Qi Tang
College of Horticulture, Hunan Agricultural University, Changsha 410128, China
E-mail: tangqi@hunau.edu.cn

Reviewer 2 Report
Comments and Suggestions for Authors
You will find some criticisms and corrections to be made. All is found in the pdf pf the ± detailed report. There are clear differences in the writing quality in different parts of the manuscript. The suggestion amde are to help the authors improve their manuscript... In blue color comments, questions etc. are indicated.

See above and in the dtailed report hopefully made available to the authors (as pdf).
Author Response

(The authors gave the same response as above.)

Round 2
Reviewer 2 Report
Comments and Suggestions for Authors
The authors have followed the advices. The revised manuscript has clearly gained in readability, mainly through condensation. Major changes concern the list of references when comparing side-by-side the first and second versions now. It is recommended to check the new list and the numbering of citations in the main text.
Comments on the Quality of English LanguageSome incomplete sentences appear in the newly edited parts (in red colour, thanks!), but this is only a minor criticism and would most likely be overlooked by potential readers of this study. There is no need to truly revise this version. The effort the corresponding author has made is obvious!